# The Evolution of Imprinted microRNAs and Their RNA Targets

**DOI:** 10.3390/genes11091038

**Published:** 2020-09-03

**Authors:** David Haig, Avantika Mainieri

**Affiliations:** Department of Organismic and Evolutionary Biology, Harvard University, 26 Oxford Street, Cambridge, MA 02138, USA; ava@asktia.com

**Keywords:** imprinting, miRNA, evolution, PTEN, IGF2, C2MC, C14MC, C19MC

## Abstract

Mammalian genomes contain many imprinted microRNAs. When an imprinted miRNA targets an unimprinted mRNA their interaction may have different fitness consequences for the loci encoding the miRNA and mRNA. In one possible outcome, the mRNA sequence evolves to evade regulation by the miRNA by a simple change of target sequence. Such a response is unavailable if the targeted sequence is strongly constrained by other functions. In these cases, the mRNA evolves to accommodate regulation by the imprinted miRNA. These evolutionary dynamics are illustrated using the examples of the imprinted C19MC cluster of miRNAs in primates and C2MC cluster in mice that are paternally expressed in placentas. The 3′ UTR of *PTEN*, a gene with growth-related and metabolic functions, appears to be an important target of miRNAs from both clusters.

## 1. Introduction

Metazoan microRNAs (miRNAs) are short RNAs (~22 nucleotides) with shorter “seed” sequences (~7 nucleotides) that interact with complementary “target” sequences in the 3′ untranslated regions (3′ UTRs) of mRNAs [1]. The interaction of seed and target commonly represses translation and accelerates decay of the mRNA [2], but other effects have been reported including prolongation of mRNA half-life [3], enhancement of translation [4], degradation of the miRNA [5], and binding at gene promoters to regulate transcription [6,7]. Each vertebrate miRNA targets, on average, two hundred mRNAs [8], and most mRNAs are targeted by one or more miRNAs [9]. Some miRNAs are evolutionarily ancient, others evolutionarily recent. Many mammalian miRNAs are imprinted, including the densest clusters of miRNAs in the genome [10].

This paper will explore the evolution of imprinted miRNAs and their target sequences, placed in the context of the evolution of unimprinted miRNAs and their targets. Section 2 discusses the evolution of miRNA–target interactions when both loci are unimprinted. Section 3 considers the more complex selective forces acting on imprinted miRNAs and their targets. Section 4 illustrates these ideas using the example of ancient miRNAs containing the hexanucleotide AAGUGC. Section 5 considers miRNAs from two recently-evolved clusters of imprinted miRNAs, the C19MC cluster of primates [11] and the C2MC cluster of mice [12]. Some miRNAs from both clusters possess the AAGUGC motif. Section 6 presents general conclusions.

A review of terminology may be helpful for some readers. The generation of a miRNA involves multiple steps from transcription of a primary miRNA precursor (pri-miRNA), to nuclear excision of the pre-miRNA hairpin from the pri-miRNA by Drosha, to cytoplasmic release of the miRNA from the pre-miRNA by Dicer. miRNAs then associate with Argonaute proteins to form the ribonucleoprotein RISC complexes that interact with target mRNAs [1]. Mature miRNAs can be derived from either the 5p or 3p arms of a pre-miRNA hairpin or from both arms.

## 2. The Co-Evolution of Seed and Target Sequences

In terms of evolutionary origins, a functional miRNA–target interaction could arise because a miRNA evolved to target an existing sequence (the miRNA is younger than the target) or because the target sequence evolved to respond to the miRNA (the target is younger than the miRNA). We will first consider the evolution of a new miRNA with a unique seed. In this case, the miRNA and its seed are younger than its potential targets. Such a seed is expected to bind to many RNAs purely by chance and the gene that encodes the miRNA (what one might call its miDNA) will increase in frequency by natural selection only if the net effect of all these interactions increases the allelic fitness of the miDNA. It seems intuitively plausible that a new miRNA would have a primary target, or small number of targets, whose regulation provided the selective advantage responsible for the miDNA’s initial increase in frequency, as well as selectively neutral or maladaptive “coincidental” targets. Subsequent selection of variants at target sites where miRNA binding to the mRNA is maladaptive would eliminate complementarity to the seed. Conversely, new targets would be recruited by natural selection when an mRNA’s response to targeting by the miRNA enhances fitness.

The likelihood that any new seed would target many mRNAs purely by chance has been considered a barrier to the establishment of new miDNAs. Chen and Rajewsky [13] and Lu et al. [14] proposed this barrier could be crossed if new miRNAs were initially expressed weakly, in a few tissues, until natural selection eliminated targets with maladaptive effects. Only then could the miRNA become a robust regulator of its adaptive targets. We question whether selection must initially be weak. A new miRNA would be less likely to cause its miDNA to increase in frequency if the miRNA’s positive effects were weak. Moreover, the weaker the negative effects of a new miRNA, or the lower the miDNA’s frequency in the gene pool, the less effective would be the selective elimination of maladaptive targets. Therefore, we suggest an alternative pathway for the successful integration of a miRNA with a new seed into gene regulatory networks. Under this scenario, the miRNA is initially subject to strong positive selection for its effects on one or a few targets, with weaker negative selection on interactions with other (coincidental) targets. Subsequent refinement of the target repertoire would occur by weaker selection on targeted mRNAs. In this process of refinement, natural selection would eliminate maladaptive targets and recruit new adaptive targets by changes to target sequences, rather than seed sequences, because a change to a seed affects the translation of multiple mRNAs whereas a change to a target directly affects the translation of a single mRNA.

The introduction of a new miRNA is somewhat simpler if the miRNA possesses an “old” seed, shared with existing miRNAs. In this case, there is an existing regulatory network of mRNAs that responds coordinately to miRNAs with this seed and 3′ UTRs have already been purged of maladaptive targets. The new miRNA is favored by natural selection if it adaptively “tweaks” the responses of the network.

The one-to-many relationship of a miRNA and its targets means that mutational changes to seed sequences will generally have larger fitness effects than mutational changes to target sequences [15]. This consideration predicts conservation of seed sequences over evolutionary time with turnover in the mRNAs to which each miRNA binds. Seed sequences of ancient miRNAs are indeed evolutionarily conserved [16] whereas the mRNAs regulated by these miRNAs show substantial evolutionary turnover [15,17,18]. Most changes to seed sequences of miRNAs that are well-integrated into regulatory networks will be rapidly eliminated by natural selection because each miRNA has multiple adaptive targets. Target sequences, by contrast, are more permissive of evolutionary change because a mutational change of a target sequence directly affects the expression of a single mRNA. The selective elimination of maladaptive targets results in 3′ UTRs that are relatively impoverished for sequences complementary to the seeds of cotranscribed miRNAs. For example, miRNA targets are depleted from *Drosophila* 3′ UTRs [19], and the 3′ UTRs of mammalian tissue-specific genes are preferentially depleted for target sequences of co-expressed miRNAs [20].

The greater evolutionary flux of mRNA target sequences than miRNA seed sequences means that mRNAs evolve to use information conveyed by miRNAs. Cellular responses to a miRNA are dictated by the evolution of targeted sequences, and by the evolution of stimuli that mediate the miRNA’s release, not by the evolution of the miRNA sequence. We have likened a miRNA to a “news report”, or “tweet”, broadcast to the cell about the conditions that led to the miRNA’s release [21]. In this analogy, a target sequence is an mRNA’s subscription to the news service. Over evolutionary time, individual mRNAs either subscribe to the service or drop their subscriptions as the information provided becomes valuable or ceases to be valuable in the control of translation. We conjecture that 3′ UTRs function as information-processing devices that “read” the state of the cell in their interactions with other RNAs and RNA-binding proteins. Just as the meaning of a word depends on its context within a sentence, the interpretation of a miRNA by a mRNA will depend on the context of the mRNA’s interactions with other RNAs and RNA-binding proteins.

## 3. The Evolution of Imprinted miRNAs

The previous section assumed a strictly cooperative interaction in which the fitnesses of targeting and targeted loci were perfectly aligned. Under this assumption, if natural selection at a miDNA locus favors an interaction with a particular mRNA, then natural selection at the locus that encodes the mRNA also favors the interaction. Genomic imprinting however can result in conflicting selective forces acting at different loci within a single genome because of effects on non-descendant kin: (i) Alleles at an unimprinted locus make a difference both when inherited from mothers and from fathers and are therefore selected to increase average inclusive fitness; (ii) alleles at a paternally-expressed locus make a difference only when inherited from fathers and are thus selected to increase patrilineal inclusive fitness; and (iii) alleles at a maternally-expressed locus make a difference only when inherited from mothers and are selected to increase matrilineal inclusive fitness [22]. Therefore, an alignment of the allelic fitnesses of targeting and targeted loci cannot be assumed if one locus is imprinted and the other unimprinted (or if the loci are oppositely imprinted).

An evolutionary conflict exists when natural selection at a protein-coding locus favors higher levels of production of the encoded protein than natural selection at a miDNA locus. Such conflicts can arise when one locus is imprinted and the other unimprinted (or when the loci are oppositely imprinted). Two scenarios can be envisioned: Evasion and accommodation. Under the evasion scenario, alleles at a targeted locus are favored that lack the target sequence and thereby evade regulation by the miRNA. As a consequence, the miRNA loses its function and becomes subject to mutational decay. Under the accommodation scenario, alleles are favored at the targeted locus that raise effective levels of gene product despite continued regulation by the miRNA.

Accommodation is the expected outcome if the targeted sequence is evolutionarily constrained to perform some important function and therefore cannot evade regulation by the miRNA simply by change to the target sequence. A simple example of accommodation would be an increase in the level of transcription of an unimprinted mRNA to compensate for inhibition of translation by an imprinted miRNA. The miRNA would now be “tethered” to its target because changes to its seed sequence would result in further increases in translated protein. The evolutionary gain or loss of other targets of a tethered miRNA would then be determined by whether the gain or loss enhanced fitness at each targeted locus. By this process, unimprinted mRNAs would be incorporated into the regulatory network of an imprinted miRNA. Such interactions are predicted to promote the fitness of each unimprinted locus (i.e., they are predicted to increase average inclusive fitness). The system remains stable so long as the fitness effects of changes in the miRNA’s seed, summed across all targeted loci, reduce the fitness of the imprinted miDNA (i.e., reducing parental-origin-specific inclusive fitness [22]).

The evasion scenario predicts evolutionarily transient interactions between targeted and targeting sequences, whereas the accommodation scenario predicts relatively stable interactions. Many interactions involving imprinted small RNAs are probably transitory, resolved by evasion, but a subset will become incorporated into regulatory networks by processes of accommodation and these will accumulate over evolutionary time. Simple corollaries are that most “old” imprinted miRNAs are predicted to be tethered to one or more targets and that most of these miRNAs’ individual interactions with unimprinted mRNAs will enhance average inclusive fitness.

One circumstance in which an unimprinted mRNA might be unable to “evade” regulation by an imprinted miRNA would arise if the targeted sequence was already regulated by existing miRNAs and a new imprinted miRNA arose with the same seed. Thus, a new miRNA with an old seed would coordinately regulate multiple members of an existing gene regulatory network. If interactions with older miRNAs with the same seed performed essential functions, then target sequences would be evolutionarily “tethered” by these functions and unable to evade an interaction with the imprinted miRNA without loss of these functions. In this case, unimprinted targeted sequences would evolve to “accommodate” their dual regulation by imprinted and unimprinted miRNAs.

## 4. Unimprinted miRNAs with an AAGUGC Motif

The hexanucleotide AAGUGC occurs in the seed sequences of several miRNAs that are highly expressed in embryonic stem cells and cancers [23,24,25,26]. These include ancient vertebrate miRNAs of the “miR-17 family” with AAAGUGC seeds, and of the “miR-302 family” with AAGUGCU seeds. These miRNA families regulate partially overlapping sets of mRNAs, in part because the octanucleotide AAAGUGCU is recognized by miRNAs of both families. Other, more recently evolved, miRNAs have converged on the AAGUGC motif, including some imprinted miRNAs from the C19MC and C2MC clusters that are therefore “new miRNAs with old seeds”. This section reviews the evolutionary history of miRNAs with the AAGUGC motif. Some possible targets of the imprinted miRNAs are discussed in Section 4.

### 4.1. miR-17, miR-20, miR-93, and miR-106

Six human miRNAs with the heptanucleotide seed AAAGUGC are processed from three pri-miRNAs: miR-17-5p and miR-20a-5p from *pri-miR-17~92* on chromosome 13; miR-93-5p and miR-106b-5p from *pri-miR-106b~25* on chromosome 7; miR-20b-5p and miR-106a-5p from *pri-miR-106a~363* on the X chromosome [26]. These are the evolutionarily oldest vertebrate miRNAs with AAAGUGC seeds. Each of their pri-miRNAs also encodes miRNAs without the AAGUGC motif: *pri-miR-17~92*, for example, encodes six pre-miRNAs, with four distinct 5p seeds, of which only miR-17-5p and miR-20a-5p possess an AAAGUGC seed. Extensive paralogy between human chromosomes 7, 13, and X suggest that these three pri-miRNAs were derived from an ancestral pri-miRNA that existed before the whole-genome duplications that occurred early in vertebrate evolution. A single “*pri-miR-17~92*” locus is reported in lampreys with multiple miRNAs with an AAAGUGC seed [27]. miR-17-related miRNAs have not been found outside of vertebrates [27,28].

The miRNAs processed from *pri-miR-17~92*, *pri-miR-106a~363,* and *pri-miR-106b~25* coordinate multiple aspects of cell proliferation by the regulation of many mRNA targets. *pri-miR-17~92*, for example, is expressed in actively-dividing mouse blastocysts, embryonic stem cells, and induced pluripotent stem cells [29,30], and is frequently amplified in human cancers [31,32]. The effects of miR-17~92 in maintaining proliferation and inhibiting differentiation of stem cells appear to oppose the pro-differentiation effects of the very ancient let-7 family of miRNAs [33]. Members of the AAAGUGC miRNA family appear to have a role in the proliferation of most stem cell populations [34], including trophoblast [35], primordial germ cells, and spermatogonia [36,37].

### 4.2. miR-302, miR-427, and miR-430

The *Xenopus* genome contains hundreds of miRNAs with AAGUGCU seeds, collectively known as miR-427-3p [38], and the zebrafish genome contains about a hundred miRNAs with AAGUGCU seeds collectively known as miR-430-3p [39]. These miRNAs are responsible for clearance of maternally-deposited mRNAs from the cytoplasm at the maternal–zygotic transition of embryonic development [40,41,42]. Six miRNAs with AAGUGCU seeds have also been reported from the lamprey *Petromyzon marinus* ([43]; variously labeled in that paper as miR-302, miR-373, miR-403).

The antisense strand of an intron of the protein-coding *LARP7* gene is transcribed as a pri-mRNA that is processed into five pre-miRNAs. Four of these pre-miRNAs (miR-302a, miR-302b, miR-302c, miR-302d) possess the heptanucleotide seed AAGUGCU at positions 2–8 of the miRNA processed from the 3p arm of the pre-miRNA. Intriguingly, miR-367-3p (processed from the fifth pre-miRNA) contains the hexanucleotide GCACUU complementary to AAGUGC. An orthologous pri-miRNA occurs in the *LARP7* genes of tetrapods (including *Xenopus*), but is absent from the *LARP7* genes of lampreys, coelacanths, and ray-finned fish [44,45]. Therefore, pri-miR-302~367 can be inferred to have originated in a common ancestor of all living tetrapods.

miR-302 has very low expression in *Xenopus* blastulas, with higher expression in somatic tissues, especially neural crest and neural cells [46]. In early chicken and mouse embryos, miR-302~367 is expressed in epiblast but not in hypoblast (primitive endoderm) [47,48]. Therefore, restriction of miR-302~367 expression to epiblast can be conjectured to be an ancestral feature of amniote development. Consistent with this conjecture, miRNAs of the miR-302~367 cluster are barely expressed in mouse embryonic stem cells but are highly expressed in epiblast-like stem cells [47,49]. However, miR-302~367 is expressed in human embryonic stem cells [50]. Although miR-302 is overexpressed in malignant germ cell tumors [51], its expression is associated with reduced proliferation and better prognosis for other tumors [52,53]. 

miRNAs with an AAGUGC seed (miR-427-3p, miR-430-3p, miR-302-3p) are often assumed to be descended from a common ancestor [54]. However, miRNAs are sufficiently short, meaning that ancestral similarity and evolutionary convergence can be difficult to distinguish. miR-302 sequences from human (hsa-miR-302), chickens (gga-miR-302c), and *Xenopus tropicalis* (xtr-miR-302) are more similar to “hsa-miR-302e” than they are to xtr-miR-427 (from *Xenopus tropicalis*) or dre-miR-430a (from zebrafish) (Figure 1). However, the similarity of “miR-302e” to the other miR-302a–d is almost certainly a result of evolutionary convergence. “miR-302e” maps to human chromosome 11, unlinked to pri-miR-302~367, and is processed from the 5p arm of its pre-miRNA, rather than the 3p arm. Orthologous sequences of Old World monkeys and apes possess AAGUGCU but the sequence is absent in platyrrhine and strepsirrhine primates. Therefore, miR-302e appears to have evolutionarily converged on the sequence of the older miR-302 family in an ancestor of catarrhine primates. By the same token, evolutionary convergence of miR-302 on the same seed as miR-427 and miR-430 cannot be rejected as an explanation of their sequence similarity. If miR-302 has a common ancestry with miR-427 and miR-430, this ancestry must be ancient because *Xenopus tropicalis* possesses many copies of miR-427 as well as the *LARP7*-associated miR-302 family.

### 4.3. miR-290~295 and miR-371~373

A cluster of miRNAs immediately adjacent to the mammal-specific *NLRP12* gene encodes multiple pre-miRNAs with the AAGUGC motif on the 3p arm. This cluster is known as miR-290~295 in mice and miR-371~373 in primates [55,56]. One human miRNA (miR-373-3p) and three murine miRNAs (miR-291-3p, miR-294-3p, miR-295-3p) have the same AAGUGCU seed as miR-302. Although the cluster is said to be restricted to eutherian mammals [56], a cluster of miRNAs adjacent to *NLRP12* in the opossum *Monodelphis domesticus* contains a miRNA with the AAGUGC sequence (miR-1643b-3p). A paternally-expressed, maternally-silent transcript antisense to miR-371~373 has been reported in human pluripotent stem cells [57].

The miR-290 cluster is broadly expressed in early mouse embryos and extraembryonic membranes, but its expression is switched off in epiblast before gastrulation [47,58]. Among post-gastrulation derivatives of epiblast, expression of miR-290~295 is restricted to the germline [58]. Thus, in gastrulating mouse embryos, the miR-302 cluster and miR-290 cluster have complementary expression domains with expression of the evolutionarily older miR-302 cluster restricted to embryonic tissues and the evolutionarily younger miR-290 cluster expressed in the germline and extraembryonic tissues. A few mice with homozygous deletion of the miR-290 cluster survive to birth but embryos with double-knockout of miR-290 and miR-302 clusters die earlier than mice with a knockout of the miR-302 cluster alone [59].

miRNAs of the miR-290 cluster appear to promote trophoblast proliferation and germ-cell differentiation in mice because homozygous deletion of the cluster results in reduced numbers of primordial germ cells [60] and reduced proliferation of trophoblast progenitors [58]. The orthologous miR-371~373 cluster probably has similar functions in human development because miR-371~373 is expressed in trophoblast [61], miRNA-372 promotes differentiation of human embryonic stem cells into primordial germ cells [62], and the cluster is highly expressed in germ cell tumors [63].

The miR-290 cluster of mice and miR-371 cluster of humans have different names because of substantial evolutionary divergence in miRNA sequences by contrast to the highly conservative nature of the evolutionarily older miR-302 cluster. This evolutionary lability has been noted by others [55,56,64]. The origin of the miR-290 cluster has been suggested to be intimately involved with the evolution of the mammalian placenta [47,59]. The evolutionary lability of this miRNA cluster matches the evolutionarily lability of placental development in eutherian mammals.

## 5. Large Clusters of Imprinted miRNAs

The majority of imprinted miRNAs are found in three large clusters labeled C2MC, C14MC, and C19MC that have prominent expression in the placenta [65]. The oldest of these clusters, C14MC, is maternally expressed and present in all eutherian mammals [66]. We have previously analyzed the unusual case of maternally-expressed miRNAs from the cluster that are antisense to the paternally-expressed *RTL1* protein-coding gene [67]. We have not undertaken a detailed evolutionary analysis of other C14MC miRNAs, but *HMGA2* is an attractive candidate to have been one of the original targets because its 3′ UTR contains multiple predicted target sites for C14MC miRNAs, with some of these target sites evolutionarily older than the C14MC cluster.

We will focus on the evolutionarily more recent C2MC and C19MC clusters. Both clusters are paternally expressed in the placenta and both contain miRNAs with an AAGUGC seed motif. Malnou and colleagues [65] have noted that the maternally-expressed C14MC cluster inhibits placental growth whereas the paternally-expressed C2MC cluster promotes placental growth, consistent with general predictions of the kin-conflict theory of the evolution of genomic imprinting. Intriguingly, paternally-expressed C19MC miRNAs are upregulated in placentas from preeclamptic pregnancies while maternally-expressed C14MC miRNAs are downregulated [68].

### 5.1. C19MC miRNAs

A large cluster of imprinted pre-miRNAs, known as C19MC, is located immediately adjacent to the miR-371~373 locus on human chromosome 19. This cluster is restricted to primates and exhibits paternal-specific expression in trophoblast. At least 22 of its members possess an AAGUGC motif on the 3p arm [69,70]. These miRNAs are processed from introns of a long non-coding RNA that is predominantly expressed from the paternally-derived chromosome in the human placenta [11,71]. Some members of the cluster produce miRNAs from both arms of the pre-miRNA, and for some pre-miRNAs the 5p arm appears to be the principal functional product. The cluster probably originated from duplications of one or more miRNAs of the adjacent miR-371~373 locus.

A new miRNA with an “old” seed coordinately regulates multiple mRNAs that are already adapted to regulation by older miRNAs with the same seed. The ancestral seed of C19MC miRNAs-3p was probably AAGUGCU (shared with miR-302-3p, miR-372-3p, miR-373-3p). This is the seed of five C19MC miRNA-3ps (miR-520a,b,c,d,e-3p) and of the chromosomally adjacent miR-372-3p and miR-373-3p. We conjecture that the C19MC miRNAs originated from duplications of pre-miR-372 or pre-miR-373 and, from their origin, regulated the “AAGUGCU network” of mRNAs. Two C19MC miRNAs (miR-519d-3p, miR-526b-3p) share an AAAGUGC seed with the ancient miR-17-5p family and are likely to regulate many of the same transcripts as the older miRNAs [72]. pre-miR-519d and pre-miR-526b are not each other’s closest relatives within the C19MC cluster and appear to have independently evolved an AAAGUGC seed on their 3p arm by a simple one-nucleotide seed-shift from AAGUGCU. *pri-miR-17~92* is expressed in proliferating trophoblast and its expression decreases with differentiation into syncytiotrophoblast [35]. The C19MC miRNAs are also expressed in proliferating trophoblast [73]. Therefore, the “AAAGUGC regulatory network” will be subject to joint regulation in trophoblast by the older unimprinted miRNAs and newer imprinted miRNAs which can be considered interjections into an existing conversation.

### 5.2. C2MC miRNAs

The C2MC cluster [12] of imprinted pre-miRNAs are transcribed from intron 10 of the mouse *Sfmbt2* locus [74,75]. C2MC miRNAs are paternally expressed in the placenta [76]. The original pre-miRNA of the cluster appears to have arisen by complementarity between (AC)_n_ and (GT)_n_ microsatellites forming two arms of a hairpin. This pre-miRNA was then duplicated several times with evolutionary divergence of seed sequences [12]. By one count, the cluster contains 36 distinct miRNAs with 23 distinct seeds including multiple miRNAs with the AAGUGC motif on the 5p arm [77]. This count does not reflect the number of pre-miRNAs because some miRNAs are present in multiple copies identical by sequence (NCBI lists 12 copies of miR-669a and 10 copies of miR-467a). The total number of pre-miRNAs per haploid genome is about 70, many producing miRNAs from both the 5p and 3p arms of the pre-miRNA.

Murine miRNAs with the AAGUGC motif, collectively known as miRNA-467-5p, evolved from miRNAs without the motif [77] and thus provide another example of recent evolutionary convergence on this ancient motif. None of these miRNAs have the full heptanucleotide seed AAAGUGC of the miR-17 family or AAGUGCU of the miR-302 family.

A repeated miRNA-encoding region in intron 10 of the *Sfmbt2* gene appears to be absent in kangaroo rats, ground squirrels, guinea pigs, and rabbits. Intron 10 of the rat *Sfmbt2* locus contains a repeated region which gives rise to a few miRNAs, none with the AAGUGC motif [12]. Thus, the C2MC cluster appears to have originated in a common ancestor of rats and mice after this lineage had diverged from other major groups of rodents.

### 5.3. Targeting of Igf2 and PTEN by C2MC and C19MC Imprinted miRNAs

The C2MC and C19MC clusters contain a mixture of miRNAs with “old” and “new” seeds. New miRNAs with old seeds target already integrated regulatory networks whereas miRNAs are expected to initially have had a small number, possibly one, adaptive target and many non-adaptive targets. As these miRNAs become an integrated part of the cellular milieu, maladaptive targets will be eliminated, and other mRNAs will evolve adaptive responses to the miRNA. We will focus on two probable targets of imprinted miRNAs from these clusters, *IGF2* and *PTEN*. The paternally-expressed growth factor IGF2 promotes trophoblast proliferation, in part by activation of the PI3K/AKT pathway, and these actions are inhibited by PTEN [78,79]. Therefore, the a priori expectation would be that a paternally-expressed miRNA would promote the expression of *IGF2* but inhibit the expression of *PTEN*. Section 5.3.1 discusses complementarity between C2MC miRNAs and the 3′ UTR of murine *Igf2*. Section 5.3.2 presents sequence-based evidence for the regulation of *PTEN* by both C2MC and C19MC miRNAs.

#### 5.3.1. Complementarity between C2MC miRNAs and Igf2 mRNA

An AC-rich sequence is present in all eutherian *IGF2* mRNAs (and can be recognized in the highly-divergent 3′ UTRs of marsupial *IGF2* mRNAs). The sequence has diverged markedly among eutherian species: For example, the heptanucleotide ACGCACA (canonical target of miR-210-3p) occurs 11 times in the AC-rich element of human *IGF2*, 12 times in the corresponding element of rhesus macaque, and once in that of the common marmoset. As another example, the heptanucleotide GCAUACA (canonical target of miR-675-3p, processed from the *H19* maternally-expressed imprinted RNA) is absent in the human element, appears six times in the macaque element, and once in the marmoset element. Predicted targets of miR-675-3p occur at similar locations in the *IGF2* 3′ UTRs of flying lemurs, thirteen-lined ground squirrels, rats, and pigs, but are absent from the corresponding sequences of mice, naked mole-rats, cattle, and hyraxes. The AC-rich sequence is polymorphic in length in the human population [80]. The functions of the AC-rich element are unknown. One possibility is that the element is a binding site for IGF2 mRNA-binding proteins (IGF2BPs) which recognize AC-rich sequences [81].

TargetScan [82] predicts target sites for several C2MC miRNAs-5p in the AC-rich sequence of the *Igf2* 3′ UTR of mice (Figure 2). We conjecture that this AC-rich sequence was the original target of the GU-rich ancestral miRNA-5p of the C2MC cluster. Of particular evolutionary significance, multiple miRNAs from mouse C2MC exhibit extended complementarity to the AC-rich element of the *Igf2* 3′ UTR of ground squirrels. This means that the first C2MC miRNAs-5p of muroid rodents would have targeted an already existing sequence in the *Igf2* 3′ UTR because ground squirrels diverged from mice and rats before the origin of the miRNAs. The consequences of binding of C2MC miRNAs-5p to the AC-rich element are unknown but we predict that these miRNAs act to promote, rather than inhibit, the translation of IGF2. The rationale for this prediction is the expectation that paternally-expressed imprinted transcripts, such as *Igf2* mRNA and the C2MC miRNAs-5p, will act coordinately to promote placental growth.

#### 5.3.2. Complementarity of C2MC and C19MC miRNAs with PTEN mRNA

PTEN is negatively regulated by miR-17-5p in some cellular contexts [83]. GCACUUU, the canonical target of miR-17-5p, appears once in the 3′ UTR of the human *PTEN* mRNA, located between two deeply-conserved polyadenylation signals (Figure 3a) [84]. This sequence participates in formation of a stem–loop structure in non-activated T cells that is bound by the RNA-binding protein roquin, obstructing binding of miR-17-5p to GCACUUU (Figure 3b) [83]. We will call this structure the “roquin hairpin”. The same sequence is also targeted by the BHRF1-3 microRNA of Epstein–Barr virus to reduce PTEN translation and thereby promote the proliferation of infected B cells [85]. Thus, multiple lines of evidence suggest that this sequence plays a key regulatory role in the control of PTEN translation.

We have explored possible foldings of this region of the *PTEN* 3′ UTR using mFold [86] and find multiple structures with roughly similar free energies. Many of these structures contain the “roquin hairpin” and another stem–loop structure 60 nucleotides in the 5′ direction we will call the “upstream hairpin”. The 38-nucleotide sequence of the “upstream hairpin” is identical in human and mouse *PTEN* genes (Figure 3c). Its loop contains GUGUCAU contiguous with UGUAGCU, respective targets of miR-425-5p and miR-221-3p. Experimental evidence exists for both miRNAs binding at this location [87,88]. A third structure predicted by mFold contains neither the “roquin hairpin” nor the “upstream hairpin”. In this structure, GUGUCAU of the “upstream” sequence pairs with AUGGCAC overlapping the miR-17-5p target sequence. We hypothesize that GUGUCAU, from the upstream hairpin, forms a pseudoknot by pairing with AUGGCAC (Figure 3d). This pseudoknot would prevent formation of the roquin hairpin and obstruct binding of miR-17-5p, and other miRNAs, to the GCACUUU target sequence. (mFold does not model pseudoknots and therefore cannot simultaneously predict formation of the upstream hairpin and pairing of GUGUCAU with AUGGCAC.)

TargetScan predicts several C2MC miRNAs-3p target the GU-rich ‘upstream hairpin’ of the *Pten* 3′UTR (Figure 4a). Sequence complementarity of these miRNAs to their target extends across the GUGUCAU sequence that is predicted to pair with AUGGCAC occluding the miR-17-5p target site (Figure 3a). We hypothesize that binding of C2MC miRNAs prevents formation of the pseudoknot, favoring the “roquin hairpin” and facilitating targeting of GCACUUU by miR-17-5p and other miRNAs with similar seeds. We also conjecture that this highly conserved sequence of the *Pten* 3′UTR was an original functional target of the C2MC cluster.

miRNAs from both the C2MC and C19MC clusters are predicted to bind at, or overlap the miR-17-5p target site. Two miRNAs of the C19MC cluster, miR-519d-3p and miR-526d-3p, are predicted to target GCACUUU (Figure 4b) in trophoblast and the former has been shown to target this site in hepatocellular carcinomas [89]. Thus, placenta-specific regulation of PTEN by miR-519d-3p and miR-526d-3p would be an example of regulation of an “old” target by “new” miRNAs. miR-467a-5p of the C2MC cluster is predicted to bind to the miR-17-5p target site with a one-nucleotide shift of target (GGCACUU). miRNAs-467 are the most abundantly expressed miRNAs of the cluster [77] with pre-miR-467a expressed from 10 tandem copies. pre-miR-467a is processed to form both miR-467a-5p, which targets the “roquin hairpin”, and miR-467a-3p which the “upstream hairpin” (Figure 4c). We conjecture that miR-467a-5p and miR-467a-3p act in concert to suppress translation of PTEN protein and that this was the principal selective reason for the ten-fold genomic amplification of pre-miR-467a. In one scenario, binding of miR-467a-5p to the “upstream hairpin” prevents formation of the pseudoknot, freeing GGCACUU for interaction with miR-467a-3p. Many other scenarios could be considered, including unfolding of pre-miR-467a, before processing by Dicer, to bind to both 5p and 3p targets.

miR-1323-5p, from the C19MC cluster, is one of the most highly expressed miRNAs in human trophoblast [90]. The AAGUGC motif is preserved on the 3p arm of pre-miR-1323 in a non-seed location but the 3p arm does not appear to produce a functional miRNA. The seed of miR-1323-5p, CAAAACU, is not shared with older miRNAs (although it is shared with the evolutionary recent miR-548o-3p). miR-1323-5p thus appears to be a miRNA with a “new” seed. The selective factor favoring the fixation of miR-1323-5p was probably its regulation of a single target, or small number of targets, with most of its interactions with mRNAs non-adaptive. TargetScan predicts more than five thousand targets of miR-1323-5p in the human transcriptome, including five sites in the human *PTEN* 3′ UTR (AGUUUUG and GUUUUGA). 

PTEN is an attractive candidate to have been one of the original functional targets of miR-1323-5p because of its important role in control of cellular proliferation and because its 3′ UTR is also targeted by other miRNAs of the C2MC and C19MC clusters. Of particular interest is the two-fold targeting of a tandem AGUUUUGC repeat that is conserved from chicken (*Gallus*) to chimpanzee (*Pan*) (Figure 5a). Intriguingly, the first of these repeats has been modified to AAUUUUGC in modern humans (Figure 5a), Neanderthals, and Denisovans (Swapan Mallick, *pers. comm*.). This sequence of the *PTEN* 3′ UTR is an anciently-conserved target site for miR-19-3p (Figure 5b). Because of the repetition of AGUUUUGC, this short sequence contains two predicted target sites for miR-1323-5p in non-human species (Figure 5c). This human-specific G-to-A substitution modifies the predicted binding of miR-19-3p and eliminates a binding site for miR-1323-5p. miR-19-3p plays a key role in suppressing PTEN in lymphomas [91]. A recent study finds that targeting of this site by miR-1323-5p increases proliferation of human esophageal cancer cells [92].

## 6. Conclusions

Oviparous mothers commit resources to offspring before fertilization. Therefore, genes expressed in embryos are unable to influence how much an embryo receives because the quantity of yolk in an egg is a *fait accompli*. By contrast, how much an embryo receives is “up for grabs” in placental development because mothers invest substantial resources after fertilization. Embryos have evolved to actively solicit or seize maternal resources and to compete with other (contemporary or future) embryos for maternal investment. This fundamental difference between prezygotic and postzygotic provisioning accounts for many distinctive features of mammalian development.

One distinctive feature is the relative timing of zygotic genome activation (ZGA). Early stages of embryonic development in *Xenopus* and zebrafish are directed by maternal gene products deposited in the oocyte with major zygotic transcription delayed until the 10th division cycle in zebrafish or twelfth division cycle in *Xenopus*. By contrast, zygotic transcription is already active in two-celled mouse and human embryos [93,94,95]. Natural selection on mammalian embryos favored early ZGA to enable embryonic genes to take an active role in resource extraction from mothers. Early ZGA created a window of opportunity in early embryogenesis, before the segregation of a dedicated germline, for the proliferation of transposable elements that depended on RNA intermediaries for their propagation [96]. Expression of these elements became a dependable part of early development.

Active participation of embryonic genes in early development created intergenerational conflicts over the allocation of maternal resources among offspring [97,98]. Embryos were selected to escalate demands on mothers who were selected to moderate embryonic demands. Because some mothers produced offspring by more than one father, genes of paternal origin in embryos were selected to impose greater demands on mothers than were genes of maternal origin [99]. This intragenomic conflict explains the importance of imprinted gene expression during mammalian development, especially in the development of placental feeding structures. By comparison, a major role of imprinted gene expression is absent during oviparous development in *Xenopus* and zebrafish because there is nothing genes of paternal origin can do to influence the amount of yolk received by their egg.

During mammalian prenatal development, genes expressed in different genetic individuals (mother and fetus) or genes of maternal and paternal origin expressed in embryos have different fitnesses with respect to transfer of maternal resources and these fitnesses cannot simultaneously be maximized by natural selection. As a consequence, early embryogenesis and placental development are evolutionarily unsettled: Adaptations of mothers to limit transfers elicit counter-adaptations of embryos to increase transfers; actions of paternally-expressed genes in embryos elicit counter-adaptations of maternally-expressed genes, and so forth. This helps to explain why early embryonic and placental development are remarkably diverse among mammalian groups. It also helps to explain why pregnancy is subject to frequent health-threatening complications because of miscommunication, or absence of communication, between mother and fetus and the weakening of homeostatic feedback [100].

A mammalian embryo’s first priority is the development of feeding structures that will be the interface for maternal–fetal exchange, with genes of paternal origin favoring relatively greater development of these structures than genes of maternal origin. Therefore, imprinted gene expression is expected to play important roles in the development of early embryos, before the segregation of extraembryonic and embryonic cell lineages, and in the proliferation and differentiation of extraembryonic stem cells. As a consequence of mother–offspring conflict and genomic imprinting, the earliest stages of embryonic development and the development of placental structures are predicted to show substantial divergence among mammalian lineages [101,102].

Girardot and colleagues [10] were puzzled by the “remarkable enrichment” of genes encoding small regulatory RNAs at imprinted chromosomal domains. We suggest a couple of factors have contributed to this enrichment. First, intragenomic conflict over levels of gene expression is predicted between oppositely imprinted genes and between imprinted and unimprinted genes. Changes in expression of genes engaged in such conflicts are thus subject to an evolutionary dynamic in which selectively-favored changes mediated by the targeting locus engender selectively-favored restorative changes at the targeted locus. Post-transcriptional regulation by small RNAs is probably easier to tinker with than transcriptional control by protein factors (six nucleotides specify a miRNA’s seed but only two amino acids). Moreover, Watson–Crick base-pairing gives small RNAs intrinsic sequence-specificity in their effects. As a consequence, small RNAs may be overrepresented among the effectors of recent regulatory changes.

Second, genomic imprinting is predicted to preferentially affect genes with dosage-sensitive effects because there is no advantage in silencing one copy of a locus if one active copy is phenotypically equivalent to two [103]. A specific prediction of the conflict model is that, at evolutionary equilibrium, the monoallelic expression of an imprinted locus is greater than the combined expression of the two alleles in related species in which the locus is unimprinted [22]. Regulatory RNAs may be particularly subject to the evolution of imprinted expression because their effects are sensitive to the relative dosage of seed and target. Although a single miRNA can catalyze multiple rounds of mRNA cleavage [104], many regulatory interactions between targeted and targeting RNAs involve one-to-one sequestration of targets [105,106,107,108]. Dosage sensitivity is also expressed in competition among different pri-miRNAs for access to the machinery of miRNA-processing [109].

Many interactions between imprinted miRNAs and unimprinted mRNAs may be evolutionarily ephemeral as target sequences evolve to evade regulation by the miRNA. If such interactions persist, then the interaction is expected either to be mutually beneficial to the targeting and targeted locus or the targeted sequence must be constrained by its performance of essential functions that would be too costly to lose. For this reason, established imprinted miRNAs are expected to preferentially target deeply conserved regulatory sites. We have previously analyzed the evolution of the interaction between one imprinted miRNA (miR-675-5p processed from the imprinted *H19* RNA) and an unimprinted mRNA (*IGF1R*). We found that the miRNA targeted the most conserved region of the entire 7 kb 3′ UTR of *IGF1R* [21]. In the present study, we have found that a deeply-conserved miR-17-5p target site in the *PTEN* 3′ UTR has been independently targeted by newly-evolved, paternally-expressed, imprinted miRNAs of the C2MC cluster in mice and C19MC cluster in humans.

In our studies of the 3′ UTRs of *PTEN* and *IGF1R*, deeply-conserved sequences are predicted to be targeted by multiple miRNAs with closely-spaced, sometimes overlapping target sequences. We interpret 3′ UTRs as information-processing devices with multiple moving parts that respond to intermolecular interactions with noncoding RNAs and RNA-binding proteins in diverse cellular milieux. For example, Mainieri and Haig [21] suggested that 3′ UTRs can “read” their cellular milieu and perform simple Boolean computations: If an effect is achieved redundantly by binding of miR-X or miR-Y, this is equivalent to X OR Y; if an effect requires binding of both miR-X and miR-Y, this is equivalent to X AND Y; if miR-Y blocks the effect of miR-X, this is equivalent to X NOT Y; and so forth.

The standard experimental protocol assesses the effects of one miRNA at a time, often by attaching a fragment of a much longer 3′ UTR to a luciferase reporter. Experiments with full-length 3′ UTRs of *IGF1R* [110] and *PTEN* [84] have shown that the experimental behavior of fragments imperfectly predicts the behavior of full-length 3′ UTRs. The standard approach of varying one miRNA at a time is not conducive to detecting the kinds of interactions among miRNAs that are predicted by the hypothesis that 3′ UTRs are computational devices. Experiments that are designed to detect complex interactions soon face a combinatorial explosion of experimental conditions (each miRNA alone, miRNAs in pairs, miRNAs in trios, etc.), especially if concentrations are also varied for each miRNA.

Comparative sequence analysis alone cannot resolve complex intramolecular and intermolecular interactions. This requires experiment. On the other hand, an experimental approach that looks at the effects of one miRNA at a time, unguided by prior hypothesis, is not conducive to the detection of complex interactions because of the large number of possible experimental conditions to be tested. There are too many degrees of freedom in both theory and experiment. Progress is likely to come from evolutionarily- and bioinformatically-informed experiments.

## Figures and Tables

**Figure 1 genes-11-01038-f001:**
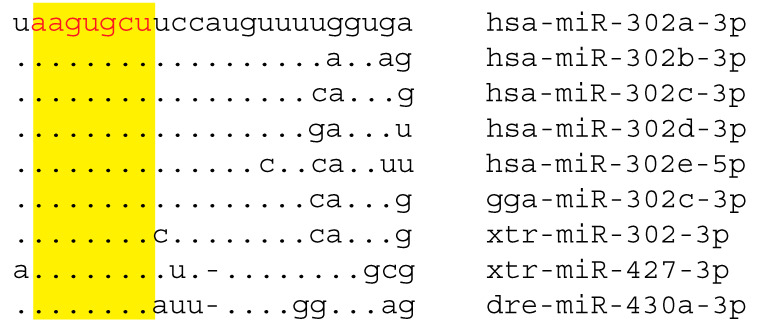
miRNAs with an AAGUGC seed. Human (*Homo sapiens*) hsa-miR-302a, b, c, d map to an intron of the *LARP7* gene and are processed from the 3p arm of their pre-miRNAs. The highly similar “hsa-miR-302e” maps to a different location and is processed from the 5p arm of its pre-miRNA. This is an example of evolutionary convergence in sequence. gga-miR-302c-3p from chicken (*Gallus gallus*) and xtr-miR-302-3p toad (*Xenopus tropicalis*) also map to the intron of *LARP7* and have a common evolutionary origin with hsa-miR-302a–d-3p. xtr-miR-427-3p from *Xenopus tropicalis* and dre-miR-430a-3p of zebrafish (*Danio rerio*) possess the same seed sequence as the miR-302s but it is unclear whether miR-427-3p and miR-430-3p are paralogous to *bona fide* miR-302s-3p or whether this is another example of evolutionary convergence. The AAGUGCU seed indicated in red font and yellow background.

**Figure 2 genes-11-01038-f002:**
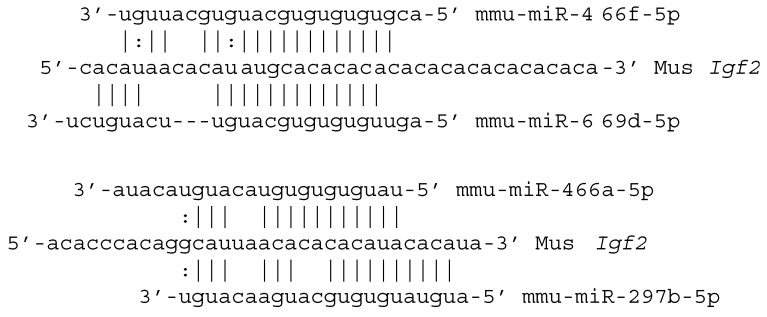
A sample of C2MC miRNAs-5p binding to the AC-rich sequence of mouse *Igf2* 3′ UTR.

**Figure 3 genes-11-01038-f003:**
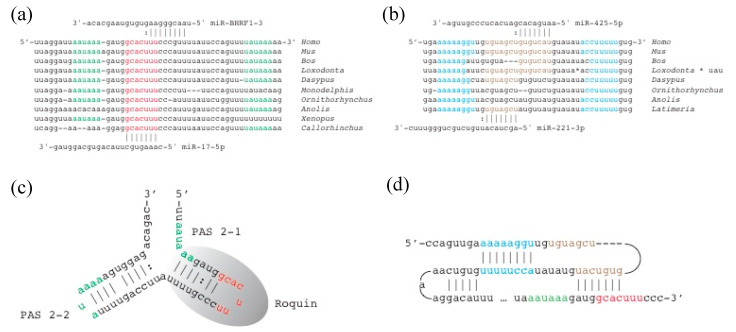
Sequence alignments and predicted structures of the *PTEN* 3′ UTR. (**a**) The sequence surrounding the “roquin hairpin” with predicted target sites of miR-17-5p and miR-BHRF1–3 also shown. (**b**) Comparative sequences of the “upstream hairpin” with predicted target sites of miR-221-3p and miR-425-5p also shown. (**c**) Structure reported in [83] with the roquin hairpin at lower right with location of polyadenylation signals. (**d**) Proposed pseudoknot in which GUGUCAU from the upstream hairpin binds to AUGGCAC, preventing formation of the roquin hairpin. In all figures, the complementary arms of the upstream hairpin in blue lettering. Polyadenylation signals in green lettering. Target sites for miR-221-3p and miR-425-5p in brown lettering. Target site for miR-17-5p in red lettering. The two sequences of (**a**,**b**) are separated by 23 highly conserved nucleotides. Common names of species: human (*Homo*), mouse (*Mus*), ox (*Bos*), elephant (*Loxodonta*), armadillo (*Dasypus*), opossum (*Monodelphis*), platypus (*Ornithorhynchus*), lizard (*Anolis*), toad (*Xenopus*), coelacanth (*Latimeria*), elephant shark (*Callorhinchus*).

**Figure 4 genes-11-01038-f004:**
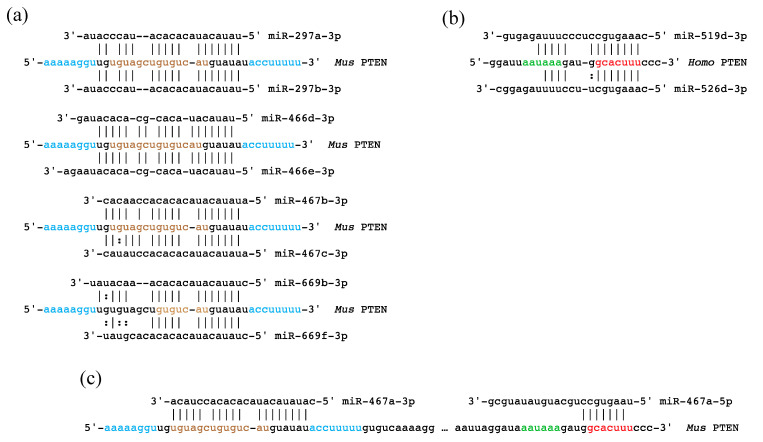
Predicted targeting of *PTEN* 3′ UTR by imprinted miRNAs. (**a**) A sample of the many miRNAs from the C2MC cluster of imprinted miRNAs that exhibit extensive complementarity to the “upstream hairpin”. (**b**) Predicted binding of miR-519d-3p and miR-526d-3p from the C19MC cluster of miRNAs to the ancient miR-17-5p target site. (**c**) pre-miR-467a is present in 10 copies per haploid genome and is processed into miR-467a-5p, which is predicted to target the miR-17-5p target site of the “roquin hairpin”, and miR-467a-3p, which is predicted to target the “upstream hairpin”. Color coding of sequence elements is the same as in Figure 3.

**Figure 5 genes-11-01038-f005:**
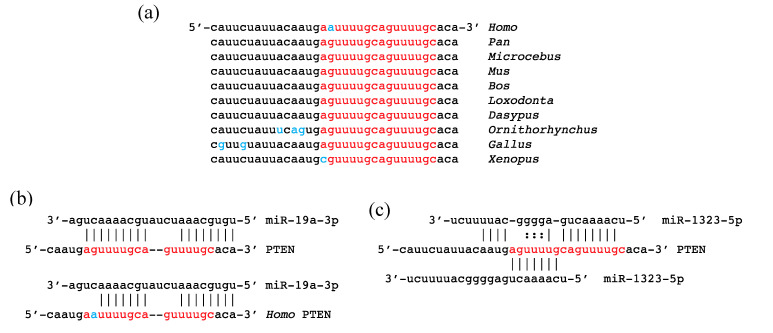
Predicted targeting of *PTEN* 3′ UTR by miR-19a-3p and miR-1323-5p. (**a**) Comparative sequence alignment showing AGUUUUGC repeat in red and differences from the consensus sequence in blue. (**b**) Targeting of the sequence by miR-19a-3p in chimpanzee (**above**) and human (**below**). (**c**) Two predicted targets for miR-1323-5p in the chimpanzee *PTEN* 3′ UTR. The lower target is lost from the human sequence. It already states "AGUUUUGC repeat in red.

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
