# Peer review of "The Evolution of Imprinted microRNAs and Their RNA Targets"

_genes, 2020, doi:10.3390/genes11091038_

Round 1

Reviewer 1 Report

Here, the authors evaluated the evolution of imprinted miRNAs in relation to their targets. Some evolution hypotheses have been postulated and some ancient clustes of miRNA have been described. Although the review is interesting, there are some concerns about its presentation. The study is hard to follow and lacks of figures which can help to understand. The cluster C19MC has been previosly described by Malnou et al, 2018, DOI: 10.3389/fgene.2018.00706 and it should be mentioned. Overall, the review try to explain the evolution equilibrium of imprinted miRNA and their target even though in some parts are described in a speculative form.

Author Response

The simplest item to deal with was reference to Malnou et al. This citation has been added. The reviewer expressed concerns about presentation and a difficulty in following the argument. We have performed a substantial revision. In particular, the introduction has been expanded to include a description of the paper's structure. I hope our speculations are useful. Science needs well formulated hypotheses as well as experiment.

Reviewer 2 Report

Overview and decision:

I applaud the authors for engaging and elucidating the idea that mammalian genomes contain imprinted microRNAs which may conflict with mRNA. The proposal that mRNA sequence may evolve to evade regulation by the miRNA by a simple change of target sequence (presuming that the underlying sequence is not heavily constrained) is a contribution. Likewise, it is an important contribution to consider the miRNA clusters in mice and primates examples of the convergent evolution to regulate a well-known tumour suppressor (i.e., PTEN). My feeling is that this paper should be published after minor revisions outlined below.

  • Please discuss the following paper discussing imprinting and miRNAs. I was surprised it was missing

MouysImprinted MicroRNA Gene Clusters in the Evolution, Development, and Functions of Mammalian Placenta by E. Cécile Malnou,1 David Umlauf,2 Maïlys Mouysset,1 and Jérôme Cavaillé2,* Front Genet. 2018; 9: 706. Published online 2019 Jan 18. doi: 10.3389/fgene.2018.00706 PMCID: PMC6346411 PMID: 30713549

  • Also, could you please discuss the following paper relating human pregnancy to miRNA?

Zhu X. M., Han T., Sargent I. L., Yin G. W., Yao Y. Q. (2009). Differential expression profile of microRNAs in human placentas from preeclamptic pregnancies vs normal pregnancies. Am. J. Obstet. Gynecol. 200, 661.e1–661.e7. 10.1016/j.ajog.2008.12.045

  • In the final paragraph (Page 11 of 20, lines 476-479), could you please expand upon long 3' UTRs are “information-processing” devices. Do you mean they are analogous to traditional neural information-processing devices via convergent evolution or do you simply mean they process information? 1-3 sentences could help expand upon this novel idea.
  • I agree that progress in this area will emerge from evolutionary-informed experiment. Very important point. To drive home this point, you state (Page 11 of 20, line 483) that experimental investigations that look at one miRNA at a time suffer a problem of “combinatorial explosion of possible conditions to be tested”. Specifically, there “are too many degrees of freedom in both theory and experiment”. This is a tad unclear, as it is not clear how looking at each miRNA is theoretically informed. Perhaps 1-2 sentences clarifying how exactly evolutionary theory can help reduce our miRNA focus and avoid the problem of “combinatorial explosion” (which you may need to define a little more).

Overall, this is important work that deserves to be published.

Author Response

Citations to the papers by Malnou et al. and Zhu et al. have been added.

The final paragraphs have been expanded with reference to our prior hypothesis that 3' UTRs can perform the equivalent of Boolean computations. The processes will be structurally distinct from neural processing. We have also added an explanation of what we meant by a combinatorial explosion. If a region of a 3' UTR is responding to five miRNAs then there are five experiments with a single miRNA, ten experiments with pairs of miRNAs, ten experiments with trios, five experiments with four miRNAs and one with all five. If dosage matters, as it probably does, then the number of treatments dramatically increases, as one independently varies the concentration of the five miRNAs.

Reviewer 3 Report

In this manuscript, Haig & Mainieri comment regarding the potential evolutionary origins and pressures of imprinted miRNAs and their targets. There are several large, recently evolved clusters of imprinted miRNAs with tissue-specific expression, but few targets of these miRNAs have been experimentally determined/verified. Using previously published experiments, seed sequence complementarity, and evolutionary analysis, the authors focus on miRNAs containing an AAGUGC motif.

They present the evolutionary history of miRNAs with this motif, and argue that some paternally-expressed miRNAs with this motif are 'new miRNAs with an old seed.' They identify PTEN as a putative target of imprinted miRNAs. PTEN is a negative regulator of IGF2 signaling, which is a paternally-expressed growth factor. Therefore, both paternally-expressed IGF2 and paternally-expressed miRNAs may regulate proliferation in a concordant manner.

Comments:

The goals and thesis of the paper should be more clearly stated in the introduction. As it stands, the lack of an overarching narrative makes the intention of specific sections unclear until the end of the paper. For example, the purpose of sections 3.1, 3.2, and 3.3 read more as a review of the literature and their purpose is not clear to the reader until some of the information becomes relevant in section 3.4. It would be helpful to cut out extraneous information, providing a brief description of the AAGUGC miRNA class and its evolutionary signficance before moving on to the more novel exploration of the regulation of PTEN and Igf2 by imprinted miRNAs.

Author Response

We have added more sign-posting of the structure of the paper but have been reluctant to eliminate 'extraneous' information that we do not find extraneous. Our interest is in the evolutionary history of seed families and of how this is affected by the intragenomic conflicts associated with genomic imprinting.

Round 2

Reviewer 1 Report

The manuscript is well structured and highly improved.

Author Response

We thank the reviewer for the stimulus to rethink the paper's organization.